# Intranasal Oxytocin and Pain Reduction: Testing a Social Cognitive Mediation Model

**DOI:** 10.3390/brainsci13121689

**Published:** 2023-12-07

**Authors:** Preston Long, Jamie L. Scholl, Xiaotian Wang, Noah A. Kallsen, Erik A. Ehli, Harry Freeman

**Affiliations:** 1Institute for Outcomes Research, Center for Medical Data Science, Medical University of Vienna, 1090 Vienna, Austria; 2Basic Biomedical Sciences & Center for Brain and Behavior Research, Sanford School of Medicine, University of South Dakota, Vermillion, SD 57069, USA; 3School of Humanities and Social Science, The Chinese University of Hong Kong (Shenzhen), Shenzhen 518172, China; xtwang@cuhk.edu.cn; 4Avera Institute for Human Genetics, Sioux Falls, SD 57105, USA; noah.kallsen@avera.org (N.A.K.); erik.ehli@avera.org (E.A.E.); 5Division of Counseling and Psychology in Education & Center for Brain and Behavior Research, School of Education, University of South Dakota, Vermillion, SD 57069, USA

**Keywords:** oxytocin, pain perception, social cognition, decision making, mediation

## Abstract

Oxytocin is well known for its role in relationships and social cognition and has more recently been implicated in pain relief and pain perception. Connections between prosocial feelings and pain relief are also well documented; however, the effects of exogenous oxytocin on social cognition and pain have not been explored. The current study tested whether intranasally delivered oxytocin affects pain perception through prosocial behaviors. Additionally, moderation of the effects of oxytocin by life history or genetic polymorphisms is examined. Young adults (*n* = 43; 65% female) were administered intranasal oxytocin (24 IU) or placebo in a crossover design on two visits separated by a one-week washout period. Pain was delivered via cold pressor. Baseline measures for decision-making and social cognition were collected, as well as pain sensitivity and medication history. Saliva samples were collected for analysis of genetic markers, and urine samples were collected to assess oxytocin saturation. Following oxytocin administration, participants reported increased prosocial cognition and decision-making. Pain perception appeared to be adaptive, with procedural order and expectation affecting perception. Finally, behavioral trust and cooperation responses were significantly predicted by genetic markers. Oxytocin may increase a patient’s trust and cooperation and reduce pain sensitivity while having fewer physiological side effects than current pharmaceutical options.

## 1. Introduction

### 1.1. Oxytocin and Pain

Oxytocin (OT) is well known for its involvement in feelings of prosocial behavior, partner selection, and its role in childbirth [1,2,3,4,5,6]. The ability of exogenous OT to mediate emotion and impulsivity has also been explored; however, less is known about OT’s ability to mediate pain. While some evidence suggests that in both rodents and humans, OT can directly alleviate pain, primarily through anti-inflammatory properties [7,8,9,10], mediation of emotional and cognitive perception of pain may be a likely mode of action in humans [11]. Additionally, OT administration may lead to adaptive pain-related behaviors such as increased trust and cooperation, particularly in acute pain conditions. The bi-dimensional relationship is also supported, with evidence demonstrating that feelings of trust and safety decrease pain ratings [12]. Furthermore, attitudinal levels of trust in an individual—an inherent predisposition to trust—may predict pain tolerance/sensitivity or pain relief efficacy. OT’s ability to amplify a sense of well-being and attenuate stress and anxiety could be the driving force behind its analgesic effect [13]. Hence, one strong theory for OT’s analgesic effect is its mediation between trust and pain, or in other words, OT attenuates pain by increasing trust. However, to date, this exploration has had mixed and inconsistent findings.

A recent meta-analysis examined the effects of intranasal oxytocin (OT_IN_) on self-perception of pain across 15 studies [14]. Results indicated that while some individual studies reported significant differences in pain perception, the overall findings yielded no significant statistical effects of OT_IN_ compared to placebo regarding subjective ratings of pain intensity or unpleasantness, pain threshold, or pain tolerance. These results challenge earlier assertions that OT directly affects pain perception in humans [15]. Alternatively, OT analgesic effects are idiosyncratic, tied to individual factors such as personality traits and life history, and to contextual factors such as situational factors and cognitive state of mind. The more complex story unfolding between OT and pain perception may be similar to the one linking oxytocin to social cognition [16].

In the remainder of this introduction, we present a review of the current research on the neurological mechanisms associating OT with pain, followed by a discussion on potential moderating factors, including life history, genetic markers, and environmental considerations. Lastly, the authors present embodied cognition as a conceptual framework from which the hypotheses on the indirect effects and context-modulated effects of OT on pain perception are formed.

### 1.2. Mechanisms

Kemp and Guastella [17] proposed a novel mechanism for explaining OT’s effects on pain based on neural systems regulating approach and avoidance motivation [17]. This is a significant theoretical advancement in the field [18]. The authors propose that the effects of OT on social behavior can be attributed to its influence on the neural systems underlying approach and avoidance motivation, suggesting that OT’s impact extends beyond social interactions to various adaptive and maladaptive behaviors. They posit that OT affects the mesocorticolimbic circuitry linked to approach and the cortico-amygdala circuitry associated with avoidance. Additionally, they contend that OT enhances the salience of personally relevant and emotionally evocative stimuli, which may apply to both social and non-social cues. The authors propose an alternative conceptual framework, the General Approach–Avoidance Oxytocin (GAAO) model, which is gaining traction in the field (see review by [18]).

Neural measurements in some studies supported the notion that OT affects pain-related neuronal circuits, but a direct relationship between neural changes and self-perception of pain was not consistently observed [18]. In conclusion, while the analysis suggests a limited direct impact of intranasal OT on self-perception of pain, the diversity and limitations of the studies warrant cautious interpretation and further research is needed in human trials to explore the potential therapeutic effects of OT_IN_ in specific painful conditions. Animal studies have consistently demonstrated that OT acts as a robust analgesic, but human studies have yielded mixed results with smaller effects [19]. This may be due to the mediating effects of the current environment during OT receipt, and past experiences with endogenous OT.

### 1.3. Life History Considerations

A dependency on context and life history may be a key explanatory factor of the varying social effects of OT_IN_. In part, this could be explained through conditioning. For instance, one of OT’s natural releases occurs during skin-on-skin contact with others, an effect that occurs independent of the type of touch or quality of the connection [20]. Thus, if a person is abused regularly in childhood, they might come to associate the release of OT with dread or disgust [21]. If true, the effects of OT would be interdependent on context and life history [16]. This lends an explanation to the effect of trust on pain, as it is quite imaginable that attitudinal levels of trust are related to life history, and suggests that life history affects the emotional valence of OT.

There is sufficient evidence to suggest an interconnection between OT and social–environmental factors. One prospective neurological investigation into the relationship between early life stress (ELS) and OT revealed that individuals who experienced emotional abuse in early life exhibit decreased functional connectivity between the right amygdala and pregenual anterior cingulate cortex (pgACC). This association remains even after accounting for age, IQ, and gray matter volume [22]. Notably, emotional abuse was the most significant predictor of this altered connectivity. Furthermore, OT’s moderating effect on this circuit was diminished in individuals with higher ELS scores, emphasizing the importance of considering both early life experiences and functional dynamics in developing OT-based therapeutic interventions [22]. Their findings provide evidence for life history’s ability to blunt the effects of OT.

Several studies have found that previous relationships with caregivers and romantic partners can shape OT’s influence, be it positively or negatively [23]. For example, OT_IN_ amplified the analgesic effect of holding a partner’s hand while experiencing pain [24]. Similarly, whether an individual is currently in a new romantic relationship or not can influence the impact of exogenous OT on feelings of attraction toward strangers [21]. In another example, the impact of OT on charitable giving was more substantial among individuals with positive experiences in their parental relationships [25]. In general, positive life history and personal familiarity tend to enhance OT effects on trust and cooperation [16].

### 1.4. Biological and Physiological Considerations

To further explore the effects of OT at the individual level, genetics must be considered. There are currently many supported single nucleotide polymorphisms (SNPs) for physiological and psychological predispositions, including response to endogenous OT. Prosocial behaviors can be influenced by OT, and polymorphisms on the OT receptor gene have been linked to trust and social cooperation [26,27]. There is mounting evidence for biomarkers associated with OT uptake and receptivity [28,29,30], where an individual’s genotype can influence sensitivity to OT responsiveness and efficacy. In the current study, we assay SNPs related to pain sensitivity, OT sensitivity, impulsivity, and attitudinal trust to determine possible relationships between pain perception and OT efficacy.

Recent research has revealed how our physical state can affect our decision-making capabilities. Factors like hunger and sleep deprivation can impact impulsivity and risk/reward behaviors [31,32]. The sensation of pain can significantly influence decision-making [33,34], as well as serve as an adaptive alarm by sending signals from the body to alert us of potential health issues and motivate us to take corrective action [35,36]. Intranasal administration of OT has been shown to enhance various prosocial behaviors such as generosity, charitable giving, empathy, and trust [16]. It has also been shown to attenuate negative drug behaviors of substances like cocaine and methamphetamine [37].

### 1.5. A Conceptual Framework

Embodied cognition is the concept that various elements present in a person’s surroundings can influence intricate human actions [38]. Studies have demonstrated that acute pain can lead to an increase in risk-taking behavior, trust, cooperation, and a likelihood to defer rewards [31,39], suggesting that our physical experiences can significantly impact our behavior. These connections between pain and decision-making can be understood as evolutionary adaptations. For example, in delayed discounting tasks, wherein a reward loses subjective value the farther out in time its receipt is anticipated, abstinence from specific drugs or a lower socioeconomic status may lead to an increased response, while deprivation from sleep, food, or water may have varying effects [40]. When it comes to managing pain, individuals tend to prioritize quick relief over delayed gratification, with chronic pain affecting the ability to see past ‘now’, including factors such as the severity of the pain, the impact it has on daily activities, and the availability of resources for long-term pain management [41]. Moreover, the level of risk associated with various pain relief methods can play a significant role in decision-making. Acute and chronic pain can drive individuals to increase risky behavior and poorer decision-making, regardless of future outcomes [33,42,43,44]. Trust and cooperation can both be considered risky and have been well established in a variety of paradigms [45,46,47,48,49,50]; however, see [51].

Thus, using OT as an analgesic may offer a unique combination of benefits under certain conditions. Correct application could potentially pave the way for a safer and more comprehensive approach to pain treatment. Traditional pharmaceutical analgesics primarily target pain reduction, but they have also been found to diminish positive social behaviors that often accompany pain. These medications, such as acetaminophen, only address pain levels and tend to diminish the prosocial effects pain has on behavior, even simple over-the-counter analgesics [52]. Therefore, a novel therapeutic approach is to leverage the prosocial and behavioral aspects of pain. OT is well situated to fill this gap. The current study measured pain-related behaviors by examining decision-making tendencies and distinct social interactions. Specifically, we investigated whether OT moderates the effects of pain on individuals’ sense of safety, trust, and willingness to cooperate. We hypothesized that acute pain would increase prosocial cognitions indicated by increased interpersonal trust, cooperation, and psychological safety scores and that the administration of OT_IN_ would directly affect these measures. Additionally, we predicted that acute pain levels would mediate the relationship between OT and decision-making, where a decreased perception of pain would be evidenced by decreased delayed discounting and loss aversion scores.

## 2. Material and Methods

### 2.1. Subject Sample and Procedure

Forty-three participants were recruited from the University of South Dakota (65% female, age M = 20, SD = 0.97). Flyers were placed around the university campus for recruitment purposes. This generated 180 pre-screened subjects. Pre-screening was conducted online, and individuals who met the criteria were invited to participate in the laboratory portion of the study, consisting of a randomized, placebo-controlled crossover study over 2 sessions, separated by a 1-week washout period (see Figure 1 for a schematic of the study procedure). A 1,2,2,1 pattern was used for randomization—the first confirmed participant received OT first, the second participant placebo, the third placebo, and the fourth OT, etc. This process was followed until near the end when necessary adjustments were made if there were more participants in one group, or cancelled appointments occurred, to rebalance the groups. The study was conducted according to the guidelines of the Declaration of Helsinki, and all participants provided written informed consent in accordance with the experimental procedures approved by the University of South Dakota Institutional Review Board (protocol IRB-18-26, approval date: 21 September 2018). Subjects were compensated $25.00 following the first visit, and $65.00 following project completion. A sample size of forty-five was targeted based on similar previous studies [53] calculated in G*Power version 3.1 using an average effect size of 0.20, with a predicted power of 0.80.

### 2.2. Laboratory Testing

Participants were screened for exclusionary criteria to determine any contraindications to intranasal oxytocin administration, including use of blood pressure, antipsychotic, or antidepressant medication, serious illness or infection, current pregnancy, or recurrent nasal or sinus pain. The most likely symptom from OT_IN_ was a possible irritated or bloody nose. Exclusionary factors were those that could influence pain scores such as current use of analgesics or recent injury. Baselines for all decision-making and social cognition measures were collected in addition to key controls such as pain sensitivity and medication history.

At arrival, participants were given verbal instructions and a sterile collection cup (BD Vacutainer, 364956, Fisher Scientific, Hanover Park, IL, USA) labeled with the participant’s specific code, time, and date to collect a urine sample. Samples were collected in a clean-catch container with a needle port and kept on ice until processing. A vacutainer was then connected to the port and aliquots were collected from the sealed system and stored at −80 °C until assayed for oxytocin. All female subjects were given a urine pregnancy test (HCG AccuMed; AccuMed Biotech LLC, Houston, TX, USA; a positive test resulted in exclusion from further participation; 0 participants were excluded) due to OT’s association with labor. A saliva sample was collected for DNA extraction and analysis (Oragene-Discover; OGR-500, DNA Genotek, Ottowa, ON, Canada).

### 2.3. Intranasal Oxytocin Administration

Participants received 24 IU of IN (van IJzendoorn and Bakermans-Kranenburg, 2015; Syntocinon^®^, NOVARTIS, Victoria Pharmacy, Zurich, Switzerland) or matched placebo (Laboswiss, Davos Platz, Switzerland) administered via nasal spray. Subjects receiving placebo at the first visit received OT at the second, for a crossover study design.

### 2.4. Measurements

Assessments began after a 30 min rest period; participants were instructed to complete an online questionnaire on decision-making, cooperation, safety, and trust, followed by the cold pressor task (detailed below). During the cold pressor task, participants underwent 6 behavioral rating tests: the Barratt Impulsivity Scale-11 [54], the Loss Aversion Scale, Delayed Discounting, the Psychological Safety Scale [55], and the Interpersonal Trust Scale [56], as well as a trust game consisting of the Prisoner’s Dilemma and the Stag Hunt. The loss aversion scale was internally developed for the study following the methods and proportions of Kahneman and Tversky’s prospect theory [57]. In addition, the delayed discounting measure was calculated using the formula presented by Kirby and Marakovic (1995) [58].

### 2.5. Barratt Impulsiveness Scale-11 (BIS-11)

The Barratt Impulsiveness Scale created by Dr. Ernest Barratt is currently in its eleventh edition. It assesses the personality trait of impulsivity across a multidimensional framework. This scale includes thirty items measuring three distinct constructs within impulsivity: attentional, motor, and nonplanning. Every item is scaled from one to four (Rarely/Never = 1, Occasionally = 2, Often = 3, Almost, Always/Always = 4). Twelve of the thirty items are reverse coded, which includes the entire subscale of self-control. Total scores on the BIS-11 are most typical; however, second-factor scores can be appropriate. The BIS-11 has a high internal reliability of 0.776 as measured by Cronbach’s alpha. It also demonstrates an acceptable test–retest reliability of 0.80 calculated using Pearson’s r [59].

### 2.6. Psychological Safety Scale

This measure assesses an individual’s perceived psychological safety within the context of a team. Psychological safety is operationalized as the degree of interpersonal risk one is comfortable with taking. Participants are asked to imagine themselves within a specific team or group and consider their responses within said context. The psychological safety scale contains seven items with a 1–5 Likert selection response. All seven items show strong inter-item correlation, and the scale has an overall high Cronbach’s alpha of 0.93 [55].

### 2.7. The Interpersonal Trust Scale (ITS)

The ITS assesses trust between people across 40 declarative statements. Twenty-five items are considered overall trust measures, and the remaining fifteen items are labeled as “filler”, meaning they are only superficially related to trust. The inclusion of these items is to boost fidelity although they are not calculated into the final score. Confirmatory Factor analysis validates the dimensions of trust and interpersonal behaviors as proposed by the author. The interpersonal Trust Scale demonstrates strong reliability and validity [56]. Research on Common Dilemmas has found this scale to be a significant predictor of trusting behavior. High scores on the ITS are correlated with an increase in the belief of promises and the degree of cooperation [60].

### 2.8. Trust Game Dilemmas

The Prisoner’s Dilemma and the Stag Hunt were used to assess applied scenarios involving trust.

The Prisoner’s Dilemma was originally created by the mathematician Albert Tucker in 1950 as a means of depicting the cost and benefits of social competition [61]. The Stag Hunt was originally created by the philosopher Jean-Jacques Rousseau. Both vignettes were adapted to follow the same mathematical format but are still applied in two different contexts. In either, there are potentially rewards and consequences for trusting others. The Prisoner’s Dilemma is set in an interrogation environment and uses sentence duration as the motivator. The length of one’s sentence depends on a combination of their choice to either remain silent or talk, and the hypothetical other’s same choice. The Stag Hunt is set in the woods and uses food as the key incentive. The Stag Hunt’s outcome is based on the player’s choice and their assumption of what the other non-real player would do. In this scenario, a player can either trust a fellow hunter to wait for the larger reward of a stag, or attempt to take a rabbit prematurely, which would provide them and them alone with food.

### 2.9. Parent Love-Withdrawal Scale

The Parent Love-Withdrawal Scale is a twenty-two-item questionnaire. It is split into two sections of repeating questions, one regarding the mother and one the father. Each item response ranges from 1 to 5 on a Likert-type scale. All of the items are negatively worded and inquire about the parent’s behavior and disciplinary approach given a variety of subject actions, such as disagreeing or disappointing. The PLWS was specifically designed and intended for use as a moderator in oxytocin studies. A meta-analysis of eight studies found consistent evidence for the association between parent–child relations and OT levels [62]. The scale has a moderate to high Cronbach’s alpha of 0.87 [25].

### 2.10. Expectancy Bias Scale

The Expectancy Bias Scale is a brief measure developed by the researchers to account for trait levels of subject optimism prior to participation. It is comprised of six items with responses ranging from 1 to 5 of likelihood that a hypothetical event would occur within the next six months. Three items were reverse coded and described negative events occurring, such as, “receiving unexpected bad news”.

### 2.11. The Visual Analog Scale

The most common subjective instrumentation of the measurement of pain is the Visual Analog Scale (VAS). It is prolific because of its simplicity and time efficiency. There are several adaptations of it including requiring participants to mark an “X” somewhere on a 10 mm line. The version used in this study scale asks for a patient’s subjective report of their pain intensity verbally on a one through ten scale. In addition, it is paired with a series of increasingly unhappy faces. The VAS has a high reliability with a Cronbach’s alpha of 0.93 [63].

### 2.12. Beck Anxiety Inventory (BAI)

The Beck Anxiety Inventory contains 21 items on a 0–3 response scale. The items are either subjective, somatic, or panic-related symptoms of anxiety. The BAI demonstrates sufficient test–retest reliability over time (r = 0.67) and a high internal consistency (Cronbach’s alpha = 0.94) [64]. The Beck Anxiety Inventory is considered the gold standard of anxiety measures within clinical psychology settings.

### 2.13. Delayed Discounting Measure (DDM)

The Delayed Discounting Measure uses the tradeoff scaling method across twelve items. Each item has two competing options. Option 1 is a monetary gain received in the present. Option 2 is a larger monetary gain received after a varying amount of days in the future. A tradeoff usually occurs from option 1 to option 2 when the participant is content with the size difference between the later reward relative to its receipt delay. This scale is meant to measure hyperbolic discounting specifically. The hyperbolic rate of discounting has already been calculated for each of the items. The hyperbolic discounting function is derived using the following formula: f(D) = 1/(1 + kt) [58]. This number describes how much an individual discounts the value of delayed rewards (when option 1 is chosen). The assigned quotient will be classified and used to mark the level of discounting the participant made at their tradeoff point. The classification for levels of discounting is broken into three categories: lowest 4 (low discounting), middle 4 (mild discounting), highest 4 (high discounting).

### 2.14. Loss Aversion Scale (LAS)

The Loss Aversion Scale measures the amount of money at which an individual is enticed enough to take a risk. The scale includes three options. Option 1 is a certainty. It yields an intermediate outcome; that is, there is no risk of loss nor chance of gain. Option 2 is a gamble yielding a gain with a probability of 50% and a loss (set at $10) with the same probability. The monetary gain is varied until indifference results and a tradeoff from certainty to risk occurs. The certain outcome is not varied and is, therefore, most naturally taken as the reference point. For a person who is averse to loss, the monetary gain must then be extra high to offset the loss [57]. The bigger the separation between the value of the gain and loss, the more loss averse an individual. The score will be calculated by dividing the expected value ((Probability of Winning) × (Amount Won per Outcome) − (Probability of Losing) × (Amount Lost per Outcome)) of the gain by the expected value of the loss. The higher the score, the more loss averse the individual.

### 2.15. Chronic Pain Grade Scale (CPGS)

The Chronic Pain Grade Scale is a short seven-item survey used to assess pain intensity and associated disability. The first three items measure pain intensity on a 1–10 scale where 0 is ‘no pain’ and 10 is ‘pain as bad as it could be’. The last four items measure pain associated with disability. The four items ask the participant to report the number of days the pain has kept them from engaging in their usual activities. The final three items use a shared 1–10 scale where 0 is ‘no change’ and 10 is ‘extreme change’ inquiring about specific activities. All items ask the participant to consider the past six months when answering [65]. All items are scored on an 11-point Likert scale, ranging from 0 to 10. The scores are calculated for 3 subscales: the characteristic pain intensity score (items 1, 2, 3) is calculated as the mean intensity ratings for reported current, worst, and average pain. The disability score (items 5, 6, 7) is calculated as the mean rating for difficulty performing daily, social, and work activities. Finally, the disability points score, which ranges from 0 to 3, is derived from a combination of ranked categories of the number of disability days (item 4) and disability score. Subscale scores for pain intensity and disability are combined to calculate a chronic pain grade that classifies chronic pain patients into 5 hierarchical categories: grades 0 (no pain) to 5 (high disability—severely limiting) [65]. Reliability and validity checks have shown this metric to be sound as seen in a Cronbach’s alpha of 0.74 and a high two-week test–retest reliability score. Higher scores on the CPGS significantly predict greater chronic pain, the likelihood of unemployment, use of opioids, physician visits, and depressed moods, all suggesting sufficient construct validity [66].

### 2.16. Cold Pressor Task

The cold pressor task was designed by a team of human factors psychologists at the Hiemstra laboratory at the University of South Dakota following the standard design [67] to maintain a precise 40-degree Fahrenheit temperature. The pressor spanned across two rooms: a storage room which contained a large freezer (set to 40 degrees) and a pump that circulated the water, and in the next room, a small cooler in which the water flowed. A small mesh lining was inserted in the cooler to guarantee the same resting depth for each inserted hand. Water was constantly recirculated back into the freezer holding a consistent temperature. A small amount of chlorine (4/ppm) was added to ensure participant safety. The pump and freezer were separated from the participation room to reduce environmental noise and distraction. All subjects were instructed to submerge their hand to the start of the wrist, at which point a five-minute timer was started. They were allowed to remove their hand if needed. Subject pain levels were recorded every 30 s at designated timepoints between the pain-relevant questionnaires. During pain exposure, the participants completed the subsection of questionnaires related to the pain hypothesis. All participants were able to complete this sub-battery within the allotted time. All pain-relevant questionnaires had their order randomized prior to completion. The examinations were all conducted in the same room. Regardless of the speed of the questionnaire completion, the participants were asked to remain in the cold pressor for the full five minutes. The researcher opened the survey page in preparation for the subjects but did not select responses on their behalf. The same researcher performed all aspects of any one subject’s trial.

### 2.17. Urinary Oxytocin Extraction

Urine samples were transported frozen on dry ice to Assay Services at the Wisconsin National Primate Center (University of Wisconsin-Madison) for OT extraction and analysis. Aliquots were thawed, vortexed, and centrifuged for solid phase extraction (SPE, 100 mg, C-18, Oasis SepPak, WAT 023590, Waters Technologies Corp, Milford, MA, USA) prior to OT analyses by enzyme-immunoassay (Oxytocin ELISA kit, ADI-901-153A, Enzo Life Sciences, Farmingdale, NY, USA). Methods for SPE of OT have been published previously [68,69,70,71,72,73]. Briefly, columns were conditioned with 1 mL methanol followed by 1 mL distilled water using vacuum pressure. Cartridges were washed with 1 mL 10% Acetonitrile/0.1% Trifluroacetic acid to waste, then eluted with 1 mL of 80% acetonitrile and evaporated in a SpeedVac at 45 °C for 1 h. Immediately after drying, samples were stored at 2–8 °C in 300 µL ethanol until assayed.

### 2.18. Urinary Oxytocin and Creatinine Analysis

Samples extracted pre- and post-OT administration were brought to room temperature and were dried and resuspended in 250 µL of Assay Buffer. Duplicates of each sample, standards, and controls were assayed following an overnight incubation. Oxytocin levels were detected using an automatic plate reader (SpectraMax, Molecular Devices, San Jose, CA, USA) at an absorbance of 405 nm. Samples were compared to known standards and absorbance values were used to determine OT values. To account for variations in concentration of urine, creatinine was measured from each sample using a picric acid microtiter plate [68,74]. Oxytocin was expressed as pg OT per mg creatinine for each sample. Pre- and post-sample and percent change were calculated for each participant. The average time between pre- and post-sample was 1:10 ± 0.00 for visit 1 and 1:04 ± 0.00 for visit 2; the non-specific binding percent was 0.09% (recommended value 0.0%). The detection limit sensitivity of the assay was 15.0 pg/mL.

### 2.19. DNA Extraction

DNA from 43 samples was extracted using the QIAsymphony^®^ instrument coupled with the QIAsymphony^®^ DSP DNA midi kit version 1.0 according to the manufacturer’s instructions (QIAGEN Inc., Germantown, MD, USA). The extracted DNA was eluted into ATE buffer composed of 10 mM Tris–Cl pH 8.3, 0.1 mM EDTA, and 0.04% NaN3 (sodium azide) with a final eluate volume of 60 μL.

### 2.20. SNP Genotyping

Extracted genomic DNA from each sample was normalized in DNA suspension buffer composed of 10 mM Tris and 0.1 mM EDTA to a working concentration of 20 ng/μL. For each SNP, normalized genomic DNA was added to a cocktail of TaqMan^®^ Genotyping Master Mix and TaqMan^®^ fluorescently labeled probes (Vic and Fam) in a ratio of 2 μL, 2.5 μL, and 1.5 μL, respectively (Thermo Fisher Scientific, Waltham, MA, USA). The DNA/PCR mix was loaded into a Bio-Rad 384-well plate (Bio-Rad, Hercules, CA, USA) to undergo PCR amplification. PCR was performed using the Applied Biosystems ViiA 7 Real-Time PCR System (Thermo Fisher Scientific, Waltham, MA, USA) with the following cycling parameters: 30 s pre-read stage at 60.0 °C, 10 min initial denaturing step at 95.0 °C, PCR stage consisting of a 92.0 °C denature step alternating with a 60.0 °C annealing step for 40 cycles, followed by a 30 s post-read stage at 60.0 °C. Genotype calls were made using TaqMan^®^ Genotyper Software v1.3.1 by Life Technologies (Thermo Fisher Scientific, Waltham, MA, USA).

### 2.21. Study Design

The treatment intervention was a double-blind placebo-controlled cross-over challenge study of OT versus placebo separated by a 1-week washout period. For the dose received, this is duration is more than adequate. After the online pre-screening procedure, the intranasal challenge was given on two visits separated by a one-week drug-free period. Subjects who met the protocol criteria were randomly allocated to receive either OT or placebo in a 1:1 ratio at the first visit. Subjects who received OT at the first visit received placebo at the second visit, and subjects that received placebo at the first visit received OT at the second visit. Subject assessments were conducted by an evaluator who was blinded to the content of the OT challenge. Immediately after OT_IN_ or placebo administration, during a rest period of 30 min, subjects were seated at a computer and completed decision-making, cooperation, safety, and trust questionnaires. Upon completion of the questionnaires, once the OT had entered the subject’s system, participants underwent a cold pressor task for five minutes. Following the cold pressor, a brief trust game was conducted, and the initial questionnaires were repeated.

## 3. Results

### 3.1. Data Exploration

All (*n* = 43) subjects completed both rounds of the trial. The data were assessed using SPSS version #27. No imputation was used for missing values and no transformations were completed following a review of the skewness and kurtosis of the primary variables. A table of the control, pain, and baseline items measuring descriptive and bivariate correlations can be seen in Table 1. A review of the table shows that parent history, expectations, and trait anxiety were all significantly correlated. Additionally, pain levels under the OT condition were significantly correlated to expectations, but pain levels under the placebo condition were not. Further exploration of bivariate correlations showed that trust game choice had a low but significant association with baseline control measures but demonstrated a weak association with independent and dependent variables. No variables were significantly correlated; however, two controls were used as covariates due to their relevance in the literature: sex (*p* = 0.059) and anxiety (*p* = 0.067), in addition to procedural order.

It was additionally inquired at the end of each laboratory visit if the subjects believed that they had received OT_IN_ or placebo. Between both visits, only 17 of 43 subjects (41%) correctly identified having received OT. Participant accuracy was consistent with previous studies that indicate little to no perceptual awareness associated with taking OT_IN_ (i.e., no better than chance).

### 3.2. Main Effects

The first set of analyses examined our hypotheses that OT would lessen pain and increase prosocial cognitions. The main effect of OT on pain was first examined. An ANCOVA was performed on pain while controlling for expectation bias (r = 0.378, *p* = 0.012) and procedural order (whether the participant received OT in the first visit or the second visit). The main effect was nonsignificant, although two associations were found with the covariates. The results also showed a significant interaction effect between procedural order and pain (F(1,41) = 5.817, *p* = 0.03, Cohen’s d = 0.112).

Mean comparisons show that subjects who received OT on their first visit (M = 5.72) reported the highest pain levels of all four groups, while subjects who received OT on visit two (M = 4.76) reported the lowest. The most painful and least painful conditions were both on OT. On the first visit, OT increased pain levels compared to placebo, but on the second visit, OT decreased pain significantly compared to placebo, illustrated by a clear moderation effect (see Figure 2).

The other covariate, expectation bias, was a significant predictor of OT pain scores (r^2^ = 0.14, *p* = 0.012), with lower expectations predicting lower pain under the OT condition, but not for placebo.

### 3.3. OT and Social Cognitions (H^1−3^)

An ANCOVA was conducted to test OT’s main effects on attitudinal trust with baseline measures and procedural order entered as covariates. There was a significant main effect of the experimental condition on attitudinal trust scores (F(1,39) = 7.011, *p* = 0.012, n*_p_*^2^ = 0.152). A comparison of the means reveals that OT increased attitudinal trust (M_PL_ = 2.61 versus M_OT_ = 2.69). A significant interaction effect was found with trust measured at baseline (F(1,39) = 4.787, *p* = 0.035, n*_p_*^2^ = 0.112), suggesting a trait bias. That is, subjects with a higher baseline level of trust experienced a greater increase in trust following OT administration. A significant interaction effect was also found with procedural order (F(1,39) = 4.085, *p* = 0.05, n*_p_*^2^ = 0.095). This moderation resulted in similar effects as seen with pain, in which the highest levels of trust were reported while on OT on visit one (M = 2.801), with the lowest levels in OT on visit two (2.571). Placebo levels of trust were practically unaffected (placebo visit 1 M = 2.619, placebo visit 2 M = 2.615).

The next ANCOVA was performed on cooperation using its baseline, procedural order, and the control variable parent love-withdrawal history (r = −0.319, *p* = 0.037) as covariates. There was a significant main effect of experimental condition on willingness to cooperate (F(1,40) = 5.819, *p* = 0.021, n*_p_*^2^ = 0.130), indicating an increase in cooperation from placebo (M_PL_ = 59.7) to drug (M_OT_ = 60.0). The interaction effects between the DV and parent love-withdrawal were nonsignificant. The interaction with baseline approached significance (F(1,40) = 3.45, *p* = 0.053) (see Table 2). Higher levels at baseline were associated with an increased effect of OT on cooperation.

### 3.4. Analysis of Covariance Mean Difference Test for Cooperation by Condition with Controls (n = 43)

The next ANCOVA was conducted to assess the difference in trust game choice between conditions. There was a significant main effect of the experimental condition on behavioral trust (F(1,40) = 9.558, *p* = 0.004, n*_p_*^2^ = 0.197). A review of the group means showed that OT exposure increased the likelihood of trust choice selection when controlling for procedural order. It should be noted that the trusting choice was more prevalent across all conditions. A significant interaction effect was not found between behavioral trust and sex or anxiety. However, post hoc analysis supports the presence of a simple effect by sex (see Table 3). A frequency comparison indicates that males were more like to switch from trusting to distrusting options under OT_IN_ while females show nonsignificant variation between conditions.

### 3.5. Frequency of Trust Decisions Made by Gender and Condition, n = 43

The final social cognition ANCOVA was conducted on safety using the baseline measure, parent love-withdrawal history (r = −0.371, *p* = 0.014), and anxiety level (r = −0.453, *p* = 0.002), and procedural order as covariates. There was a nonsignificant main effect of experimental condition on psychological safety (F(1,40) = 0.061, *p* = 0.806). No significant interaction effects were found.

*Summary.* The results largely support hypothesis two. Attitudinal trust, cooperation, and behavioral trust were directly and positively affected by the administration of OT. Furthermore, baseline moderations support the trait nature of attitudinal trust and cooperation. Their interactions support the literature, demonstrating individual characteristics’ influence on OT’s effect. While pain did not show a main effect, there was a surprising moderation of procedural order that demonstrated OT’s analgesic capacity. Only the social cognition variable safety was unaffected. The identification of the main effects was the first step toward establishing possible mediators.

### 3.6. Mediations (H^4^)

Hypothesis three examined social cognitive variables as possible mediators of the effect between OT and pain. The results of the ANCOVAs show evidence of OT’s effect on the social cognitive factors of attitudinal trust, behavioral trust, and cooperation. OT may increase prosocial feelings, specifically cooperation and attitudinal trust, but there is insufficient evidence to claim a mediation effect. Increases in cooperation and trust were not associated with changes in pain levels as indicated by bivariate correlations. Multiple regressions show trust and cooperation scores were also not predictive of pain scores; the factors vary independently.

Furthermore, changes in trust game choice were not predictive of pain levels. OT did not affect safety and therefore cannot be considered. Overall, hypothesis three is not fully supported. The ANCOVAs demonstrate that OT can increase prosocial cognitions and reduce pain, but a link between these two effects could not be fully established. However, this does not rule out pain as a possible mediator because of design limitations regarding temporal order and predicted effects between social cognitive variables and pain. This issue is more fully addressed in the discussion.

### 3.7. Decision Making and OT (H^5^)

Hypothesis five examined pain as a possible mediator between OT and decision-making, proposing that OT’s reduction in pain will influence decision outcomes. In order to indirectly assess pain as a mediator between OT and decision-making, the main effects must be present. Two one-way repeated measures ANCOVAs were conducted to determine if there was a statistically significant difference between the placebo and OT conditions for the two decision-making variables, loss aversion and self-control. The first analysis examined mean differences in loss aversion while controlling for baseline and procedural order. A significant main effect was found (F(1,37) = 4.491, *p* = 0.041, Cohen’s d = 0.108). Main effect comparisons showed that loss aversion increased from the placebo (M = 0.55) to the drug condition (M = 0.860), i.e., participants under the influence of OT became more risk averse. No significant interaction effects were found.

The final decision-making ANCOVA compared OT’s effects on the composite variable self-control, an aggregate of the z-scores for the decision-making variables. The results showed a main effect of condition on self-control (F(1,39) = 4.089, *p* = 0.05, n*_p_*^2^ = 0.095). The group means are less interpretable due to the required transformation of the composite variable; however, the placebo condition scores demonstrate increased self-control compared to the drug condition. A review of scatterplots showed an interesting relationship between pain, OT, and self-control. Pain was predictive of self-control (r^2^ = 149, *p* = 0.011) only in the OT condition (see Figure 3).

### 3.8. Post Hoc Analysis

A series of between-subject analyses in the form of t-tests were performed. Mean comparisons between gender and between intervention and placebo effects resulted in no significant differences across pain levels. T-tests were additionally run to assess the effects of OT on the primary research variables of trust, cooperation, and safety. Again, no significant differences were found.

Furthermore, the primary ANCOVAs were then rerun without controlling for procedural order. Behavioral trust and loss aversion showed a decreased effect size and significance, while attitudinal trust lost significance entirely. The baselines for cooperation and attitudinal trust also lost their significant interaction effects. The variable safety was unaffected. These findings provide evidence for the strong influence procedural order had on OT effects. Self-control was the only variable that lost significance when including procedural order in the analysis. A significant interaction effect was found with the self-control baseline, suggesting a strong trait bias (F(1,34) = 7.418, *p* = 0.10). This suggests a direct effect of OT on decision-making since its effect was consistent between visits even when OT’s relationship with pain was reversed. To increase the understanding of this relationship, the variables comprising self-control were also assessed. An ANCOVA was performed on delayed discounting and impulsivity using their respective baselines and procedural orders as covariates. OT had a significant main effect on impulsivity (F(1,40) = 6.789, *p* = 0.013, n*_p_*^2^ = 0.145). There was also a significant interaction with baseline (F(1,39) = 1.960, *p* = 0.170). A review of the group means supported the self-control finding showing elevated levels of impulsivity on OT. Delayed discounting did not have a significant main effect.

### 3.9. Exploratory Genetics

Genetic markers were evaluated as possible predictors and covariates. To begin, a correlation matrix was reviewed for all baselines and controls (see Table 4). The results showed no significant correlations between the six SNPs and any decision-making variables. No correlations were found between SNPs and pain levels. Significant correlations were found between the genetic markers and social cognitive variables. Behavioral trust scores were significantly associated with the Early Life Risk *SLC6A4 rs25531* marker (r = 0.322 *p* = 0.035). In addition, cooperation was significantly correlated with markers for pain sensitivity, impulsivity, and executive functioning.

Next, correlations between markers and control variables were considered. One significant negative correlation was found between the prosocial *OTXR rs53576* (r = −0.308 *p* = 0.047) and chronic pain. Regression analysis was conducted for each SNP with their theoretical indicators as measured in study questionnaires. No genetic markers significantly predicted their associated indicator.

For exploratory purposes, the hypothesis tests were rerun, including genetic markers as covariates. SNPs were chosen based on their relevance supported in the literature. Two of the seven ANCOVAs were significantly affected. Cooperation had a marked increase in main effect size (n*_p_*^2^ = 243) and had a new significant interaction with parent love-withdrawal history when SNPs for prosocial disposition, early life risk sensitivity, and OT sensitivity were controlled (F(1,35) = 5.236, *p* = 0.028, n*_p_*^2^ = 0.141). Lastly, the ANCOVA for loss aversion lost significance when additionally controlling for SNPs indicating OT sensitivity, impulsivity, and executive functioning. However, a new significant interaction occurred between loss aversion and the marker for impulsivity (*SCL6A3 rs460000*) (F(1,33) = 4.416, *p* = 0.043).

## 4. Discussion

The effect of OT can often appear paradoxical, as reflected in the literature. This study found similar conflicting findings, although as research continues, our understanding of these findings seems less perplexing. A primary notable finding was the reversal in pain perception between the first and second exposures, suggesting an adaptive response. During the first exposure, characterized by uncertainty and increased risk, OT_IN_ resulted in heightened pain sensitivity, loss aversion, and delayed discounting, which could be evolutionarily advantageous. However, upon the second exposure, when uncertainty was reduced, OT increased pain tolerance, providing protective benefits once significant survival risk was ruled out. When considered alongside environmental and life history factors, the seemingly inconsistent influence of OT becomes more predictable. Our study revealed a significant effect of OT on pain perception, particularly in the context of repeated exposure to pain. This provides the first evidence for the need to examine the effects of OT over time and across repeated exposures with the same individual.

Moreover, the study demonstrated that acute pain increased state trust and cooperation, effects that may be hindered by conventional analgesics. Although the observed effects were moderate, the findings suggest that OT has the potential to sustain and strengthen prosocial attributes. Therefore, OT administration could enhance patient trust, cooperation with healthcare providers, and adherence to treatment regimens. A seemingly paradoxical relationship has also been identified, in which trust decreases pain and pain increases trust. However, this may be resolved by considering the different types of trust, specifically, attitudinal/trait and state. The present study only examined the former. But, it may be that pain is lessoned by state trust, while pain increases attitudinal trust, as supported by the findings of this study. This trait/state distinction could account for the seemingly contradictory relationships between pain and social cognitions.

Lastly, exposure to OT increased risk aversion and impulsivity. This effect was consistent after controlling for procedural order. Examination of the latent variable of self-control also showed a decreased average in the experimental condition. Although, lower pain levels were associated with reduced decision-making biases overall and OT was associated with lower pain levels, as expected. However, there is evidence that OT also has a direct effect on decision-making beyond that of its effect on pain. A direct examination of the effect of OT on decision-making would lend insight into this relatively unexplored domain.

### 4.1. Limitations

For subject safety, the study employed the standard, minimum effective dose of OT reported in the literature. However, future research should explore the effects of higher dosage levels while controlling for endogenous levels. It should also be noted that the completion of questionnaires may have offered a distraction, resulting in a small pain reduction across groups. In addition, future research should be conducted to assess the possible range of effect sizes across individuals and contexts, and preferably with a larger sample size. While the primary analysis benefited from within-subject testing, the post hoc exploration of between-subject effects was likely underpowered. It remains unclear what the range of potential for OT’s analgesic effect is in comparison to familiar analgesics, such as ibuprofen, for example. An RCT should be conducted to compare efficacy across a number of common pain alleviation treatments. Furthermore, it is crucial to validate life history variables and other risk assessment measures to identify patients who are less likely to benefit from integrative health approaches involving OT.

While the mechanisms at play in the release and activation of OT provided a model of understanding for this research, these mechanisms were never directly tested with physiological measures. The OT research field would strongly benefit from more direct examinations of the neurological mechanisms and their interaction with OT to further clarify the neurocircuitry of the response. FMRI studies would help continue to tease apart the top-down and bottom-up modalities. Additionally, an appropriate next step would be to separate out the timing of the prosocial measures from the pain exposure. This allowed for the interaction to be examined, but also temporally cross-contaminated the outcomes.

### 4.2. Future Directions

To gain a clearer understanding of oxytocin’s (OT) role in pain reduction, whether as an endogenous hormone or through therapeutic application, studies incorporating mediation and moderation analysis are essential. The extensive literature on the relationship between OT and prosocial behavior warns against making direct causal claims. For instance, exogenous OT has elicited paradoxical responses in some individuals, heightening antisocial behavior among those with aggressive traits towards outgroup members [75,76,77,78,79]. Likewise, in certain scenarios, administering OT can intensify emotional and physical distancing based on individual familiarity and romantic relationship status [21,80]. Additionally, the dose–response effects of exogenous OT differ between males and females [77]. Incorporating pain, as observed in this study, introduces an added layer of complexity. Our research, which considered various situational and individual factors, revealed that OT manifested paradoxical effects on pain and prosocial behavior under specific conditions and among particular individuals. Models predicting the link between OT, pain perception, and management will undoubtedly become more refined as future research incorporates individual traits, situational contexts, gender, and life history as moderating elements. This will reduce the likelihood of unintended consequences in clinical settings. Research should also be conducted to assess the OT response over time when treating chronic pain conditions.

An additional focus for upcoming research should be to leverage advancements in imaging technology to examine the neural pathways involved in OT and pain. This method could be particularly effective in clarifying the activation pathways to and from the amygdala, a central hub for both pain modulation and social cognition. The amygdala is a direct target for oxytocin through meso-cortical circuits and is also influenced by top-down projections (e.g., the fronto-amygdala pathway). Recent studies suggest that top-down and bottom-up OT projections activate distinct amygdala subregions [81]. An example of this is how OT can prompt either avoidance or approach behaviors based on specific activation pathways [82]. Furthermore, conditions like clinical depression may disrupt the fronto-amygdala circuitry, potentially diminishing OT’s impact. Currently, these differential pathways have not been explored in studies investigating exogenous effects on pain perception. However, delving into such areas could greatly enhance our comprehension of how OT influences pain perception in varying individuals and situations.

## 5. Conclusions

The most pronounced effect was seen in the distinct shift of OT’s effect between the first and second exposures to pain. This reversal in pain perception appears to be adaptive, as seen in the decision-making variables. The first exposure involves uncertainty and, therefore, more risk, so increased pain, loss aversion, and delayed discounting would all be evolutionarily advantageous. However, upon the second exposure, when the threat of uncertainty is relieved, the OT increases pain tolerance which can then be considered protective. This phenomenon could be thought of as the pain sensitivity–tolerance tradeoff. Acute pain was shown generally to increase state trust and cooperation as is consistent with the literature. However, this positive effect may be nullified by typical analgesics. While the discovered effects were moderate, this study’s findings suggest OT is able to maintain and even reinforce these prosocial attributes. Thus, OT could increase patient trust and cooperation with staff, as well as adherence to treatment regimens. An OT-based pharmaco-behavioral therapy could likely supplement traditional psychoeducation pain programs or cognitive-behavioral therapies and demonstrates potential as a titration aid for more problematic painkillers.

## Figures and Tables

**Figure 1 brainsci-13-01689-f001:**
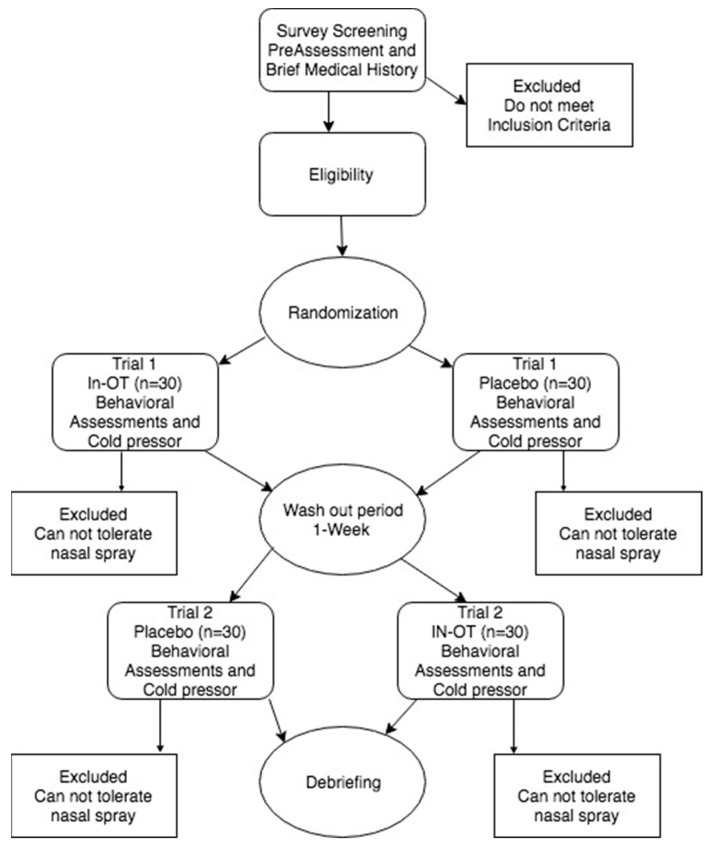
Study plan experimental concept design.

**Figure 2 brainsci-13-01689-f002:**
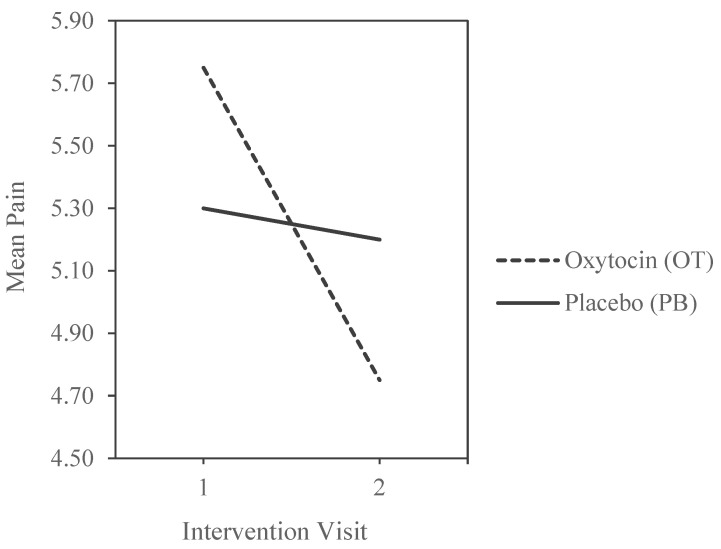
Pain scores by experimental order.

**Figure 3 brainsci-13-01689-f003:**
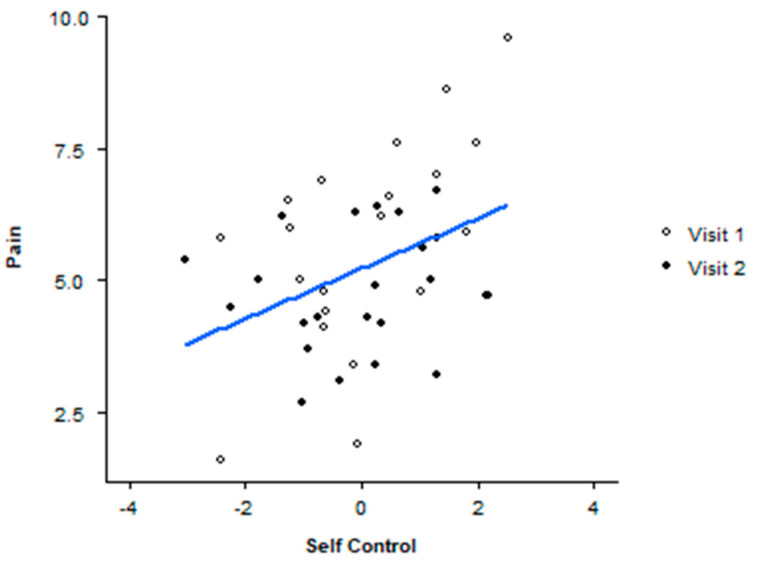
Pain score by self-control under OT condition. The slope of the regression can be seen in the blue line.

**Table 1 brainsci-13-01689-t001:** Descriptives.

	Min	Max	Mean	SD	1.	2.	3.	4.	5.	6.	7.	8.	9.	10.	11.	12.
Control Descriptives																
1. Impulsivity	1.54	3.02	2.17	0.37	–											
2. Chronic Pain	1.00	5.29	2.45	1.09	0.06	–										
3. Anxiety	0.00	44.00	15.07	11.99	**0.48** **	0.15	–									
4. Expectations	1.50	5.00	3.64	0.62	**−0.28** **	0.22	**−0.31** *	–								
5. Parent History	1.14	4.00	2.03	0.68	**0.30** **	−0.10	**0.48** **	**−0.35** *	–							
Baseline Descriptives																
6. Cooperation	44.62	90.11	61.71	10.87	−0.16	0.13	−0.13	0.16	**−0.19** *	–						
7. Safety	15.00	29.00	23.04	3.67	**−0.32** **	**0.18** *	**−0.27** **	**0.17** *	−0.17	−0.01	–					
8. Trust	1.92	3.28	2.54	0.35	**−0.19** *	−0.03	−0.06	−0.003	−0.08	0.01	0.14	–				
9. Delayed Discounting	0.00	0.03	0.02	0.01	0.08	0.10	0.09	0.08	0.05	0.08	0.02	−0.08	–			
10. Loss Aversion	0.00	2.20	0.77	0.84	0.07	−0.08	0.15	−0.09	0.09	−0.03	−0.11	0.03	0.11	–		
**Treatment**																
11. Oxytocin	1.60	9.60	5.23	1.67	0.22	0.08	−0.11	**0.38** *	−0.24	−0.12	0.12	−0.21	0.10	0.15	–	
12. Placebo	1.00	9.80	5.24	1.91	0.358	0.06	−0.01	0.21	−0.21	0.03	0.23	0.12	0.03	−0.06	**0.62** **	–

* *p* > 0.05; ** *p* > 0.01; *n* = 43. Descriptive statistics for the key variables. Correlations reported as Pearson’s r. Bold indicates significant correlation.

**Table 2 brainsci-13-01689-t002:** Repeated measures ANCOVA cooperation.

Within Subjects Effects					
	SS	df	MS	F	*p*
Cooperation	**286.282**	**1**	286.282	6.124	**0.018**
Cooperation * Baseline	180.278	1	180.278	3.856	0.053
Cooperation * Parent-Love History	161.369	1	161.369	3.452	0.071
Cooperation * Procedural Order	18.525	1	18.525	0.390	0.535
Residual	1869.896	40	46.747		

Bold indicates significant effects. * indicates an interaction.

**Table 3 brainsci-13-01689-t003:** Trust behavior by sex.

Count		Condition	
		OT	PB
Males	Trust	9	13
	Do not Trust	6	2
Females	Trust	24	23
	Do not Trust	4	5

**Table 4 brainsci-13-01689-t004:** Genetic marker associations.

Pearson Correlations					
		GMPain	GMImpuls	GMExFunc	Cooperation
GMPain	Pearson’s r	—			
	*p*-value	—			
GMImpuls	Pearson’s r	−0.143	—		
	*p*-value	0.362	—		
GMExFunc	Pearson’s r	**0.681** ***	−0.064	—	
	*p*-value	<0.001	0.681	—	
Cooperation	Pearson’s r	**−0.316** *	**0.314** *	**−0.330** *	—
	*p*-value	0.039	0.040	0.031	—

* *p* < 0.05, *** *p* < 0.001, *n* = 43. Series of correlations between SNPs and cooperation. Bold indicates significant correlation.

## Data Availability

The datasets generated for this study are available upon request to all interested researchers. The raw data supporting the conclusions of this article will be made available by the authors without undue reservation. The data are not publicly available due to a lack of obtained consent for third-party use or public disclosure.

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
