# Peer review of "Intranasal Oxytocin and Pain Reduction: Testing a Social Cognitive Mediation Model"

_brainsci, 2023, doi:10.3390/brainsci13121689_

Round 1

Reviewer 1 Report

Comments and Suggestions for Authors

The work is interesting and, importantly, confirms the known fact that oxytocin can change the perception of pain behavior  not only in labor. However, I have a few questions.

1. There was quite a large battery of neuropsychological tests that took a long time to complete. It was not indicated how much time the patient had to complete or did complete all the questionnaires and whether the time required to complete the questionnaires distracted him from the pain stimulus in paralel group.

2. The patient's level of attention was not tested.

3.

The dose of oxytocin was very small. Have other factors that may influence the perception of the pain stimulus been excluded? including: the order of completing the questionnaires, fatigue with numerous neuropsychological tests and environmental factors - location of the examination, time of completing the tests.

4. To what extent did the researcher help the respondents themselves in completing the surveys?

5. did the same researcher who performed the neuropsychological tests dose the pain stimulus and did the same researcher determine the level of pain?

These questions do not diminish the value of the work, which is interesting and innovative and seeks an important answer to the question of the extent to which oxytocin changes the perception of pain.

Author Response

  1. There was quite a large battery of neuropsychological tests that took a long time to complete. It was not indicated how much time the patient had to complete or did complete all the questionnaires and whether the time required to complete the questionnaires distracted him from the pain stimulus in parallel group.

The pain exposure lasted five minutes in which time the participants completed the subsection of questionnaires related to the pain hypothesis. All participants were able to complete this sub-battery within the allotted time. The possible distracting feature of this design will be added to the limitations sections.

  1. The patient's level of attention was not tested.

The patient’s level of attentional impulsivity was tested as a domain of the BIS-11.

  1. The dose of oxytocin was very small. Have other factors that may influence the perception of the pain stimulus been excluded? including: the order of completing the questionnaires, fatigue with numerous neuropsychological tests and environmental factors - location of the examination, time of completing the tests.

 All pain-relevant questionnaires had their order randomized prior to completion. The examinations were all conducted in the same room. Regardless of the speed of the questionnaires completion the participants were asked to remain in the cold-pressor for the full five minutes.

  1. To what extent did the researcher help the respondents themselves in completing the surveys?

The researcher opened the survey page in preparation for the subjects but did not select responses on their behalf.

  1. Did the same researcher who performed the neuropsychological tests dose the pain stimulus and did the same researcher determine the level of pain?

The same researcher performed all aspects of any particular subject’s trial.  

Reviewer 2 Report

Comments and Suggestions for Authors

The manuscript ‘Intranasal Oxytocin and Pain Reduction: Testing a Social Cognitive Mediation Model‘ analyzes the effects of exogenous oxytocin on social cognition and pain that has not been frequently explored. The study is intended to test whether intranasally-delivered oxytocin affects pain perception through prosocial behaviors and examined the moderation of the effects of oxytocin by life history or genetic polymorphisms.

Previous studies have explored the potential analgesic effects of oxytocin, and some studies have suggested that it may indeed have a role in modulating pain perception. The mechanism behind this is not fully understood, but oxytocin receptors are found in areas of the brain associated with pain processing. Intranasal administration of oxytocin has been a common method in studies investigating its effects. The submitted manuscript explores  these aspects by testing a social cognitive mediation model.

After going through the manuscript, I have following comments for the authors.

1.      Several studies have explored the impact of intranasal oxytocin on various types of pain, including experimental pain, acute pain, and chronic pain conditions. However, the results have been mixed, with some studies suggesting positive effects on pain perception, while others have not found significant effects. Was the effect of oxytocin examined in different taypes of pains in the current study?

2.      The introduction section is way too long with a lot of repeated informaion. Please trim the introduction section to make it reader-friendly.

3.      The legends of the figures are double. Please the legends above the figures.

4.      The abbreviations in the tables are not described in the legend.

Comments on the Quality of English Language

Language is fine. Minor grammatical and syntax corrections needed.

Author Response

  1. Several studies have explored the impact of intranasal oxytocin on various types of pain, including experimental pain, acute pain, and chronic pain conditions. However, the results have been mixed, with some studies suggesting positive effects on pain perception, while others have not found significant effects. Was the effect of oxytocin examined in different taypes of pains in the current study?

No, only experimental pain was directly examined. Acute pain was excluded and chronic pain was statistically controlled for.

  1. The introduction section is way too long with a lot of repeated information. Please trim the introduction section to make it reader-friendly.

Okay, we will explore areas for reduction. Thank you for the feedback.

  1. The legends of the figures are double. Please the legends above the figures.

I believe that formatting decision is made by the journal but we will reach out on this issue.

  1. The abbreviations in the tables are not described in the legend.

This will be addressed.

Reviewer 3 Report

Comments and Suggestions for Authors

Please add if one week wash out period s appropriate or can it be longer?

Please add any side effects of OT and wether it ca be gender or age related? Can anyone become hypersensitive or hyper responsive to OT whe it s used for pain?

In the autors opinon, which types of pain migth be more responive to OT, or the effect ca be independent of the type of pain, length of it or mechaism underlying the pain? Please elaborate.

Author Response

  1. Please add if one week wash out period s appropriate or can it be longer?

A one-week washout is on the longer than needed side for the dose received. This statement will be added to the paper.

  1. Please add any side effects of OT and wether it ca be gender or age related? Can anyone become hypersensitive or hyper responsive to OT whe it s used for pain?

Okay, we will include information on these topics.

  1. In the autors opinon, which types of pain migth be more responive to OT, or the effect ca be independent of the type of pain, length of it or mechaism underlying the pain? Please elaborate.

In the author’s opinion, pain responsiveness to OT would largely be independent from the pain type, whether experimental or acute. This is also believed to be the case concerning the underlying mechanisms, whether pressure or thermal, for example. There may be an altered effect on long-term/chronic pain however due to the need for repeated exposure.

Reviewer 4 Report

Comments and Suggestions for Authors

The authors of the manuscript perform several psychological tasks, including measurement of oxytocin in the urine of participants of a randomized placebo-controlled study including intranasal oxytocin administration to evaluate the role of oxytocin for pain reduction and modulation of this process by trust and genetic factors. Before publication of the manuscript, authors should address several issues.

1) Line 173 of the manuscript. What are the criteria for online pre-screening?

2) How randomization was achieved in your study?

3) Please correct Table 1, in the current format, it’s difficult to read. Increase the size of the first column and adjust the numbers.

4) How did you use data on oxytocin in urine? It’s not clear from the text

5) Also persons who have initially high oxytocin can be less sensitive to intranasal application. Can this affect your results?

Author Response

  • Line 173 of the manuscript. What are the criteria for online pre-screening?

Flyers were placed around the university campus for recruitment. This generated 180 online pre-screened subjects. 

  • How randomization was achieved in your study?

A 1,2,2,1 pattern was used for randomzation - first confirmed participant got OT first, second participant placebo, third placebo and fourth OT, etc. This process was followed until near the end when necessary adjustments were made if there were more participants in one group, or cancelled appointments occurred, to rebalance the groups.

3) Please correct Table 1, in the current format, it’s difficult to read. Increase the size of the first column and adjust the numbers.

We have done so. Thank you for the feedback.

  • How did you use data on oxytocin in urine? It’s not clear from the text

The pre-post samples of urine were used to assess the absorption rate of OT while controlling for creatinine levels. However, due to limited funds we were unable to analyze a large enough subsample for adequate power.

  • Also persons who have initially high oxytocin can be less sensitive to intranasal application. Can this affect your results?

This is possible. We will include a statement in the limitations section.

Reviewer 5 Report

Comments and Suggestions for Authors

Considering that your study, among others, is related to social cognition, some recent references could be added (Example: Voncken MJ, et al. The effect of intranasally administered oxytocin on observed social behavior in social anxiety disorder. Eur Neuropsychopharmacol 2021; 53:25). Furthermore, it is written in your manuscript that “the study employed the smallest effective dose of OT reported in the literature”. However, this dose is standard in other related older and recent articles (Examples: Guastella AJ, et al. A randomized controlled trial of intranasal oxytocin as an adjunct to exposure therapy for social anxiety disorder. Psychoneuroendocrinology 2009; 34:917. Voncken MJ, et al. The effect of intranasally administered oxytocin on observed social behavior in social anxiety disorder. Eur Neuropsychopharmacol 2021; 53:25).  

Author Response

Thank you for the references. We will explore potential places for them. We will also clarify the wording to express more accurately what was intended by, “the smallest effective dose”.